# COVID-19 Vaccination in Young People with Functional Neurological Disorder: A Case-Control Study

**DOI:** 10.3390/vaccines10122031

**Published:** 2022-11-28

**Authors:** Natalie Lim, Nicholas Wood, Archana Prasad, Karen Waters, Davinder Singh-Grewal, Russell C. Dale, Joseph Elkadi, Stephen Scher, Kasia Kozlowska

**Affiliations:** 1Department of Psychological Medicine, The Children’s Hospital at Westmead, Westmead, NSW 2145, Australia; 2National Centre for Immunisation Research and Surveillance, Kids Research, Sydney Children’s Hospitals Network, Westmead, NSW 2145, Australia; 3The Children’s Hospital at Westmead Clinical School, Faculty of Medicine and Health, University of Sydney, Westmead, NSW 2145, Australia; 4Department of General Medicine, The Children’s Hospital at Westmead Clinical School, Westmead, NSW 2145, Australia; 5Sleep Medicine, The Children’s Hospital at Westmead, Westmead Clinical School, Westmead, NSW 2145, Australia; 6Specialty of Child and Adolescent Health, Faculty of Medicine and Health, University of Sydney, Westmead, NSW 2145, Australia; 7Department of Rheumatology, Sydney Children’s Hospital Network, Westmead, NSW 2145, Australia; 8School of Women’s and Children’s Health, University of New South Wales, Randwick, NSW 2031, Australia; 9Kids Neuroscience Centre, Faculty of Medicine and Health, University of Sydney, Westmead, NSW 2145, Australia; 10The Brain and Mind Centre, University of Sydney, Camperdown, NSW 2050, Australia; 11McLean Hospital, Belmont, MA 02478, USA; 12Department of Psychiatry, Harvard Medical School, Boston, MA 02215, USA; 13Speciality of Psychiatry, Faculty of Medicine and Health, University of Sydney, Westmead, NSW 2145, Australia; 14Brain Dynamics Centre at Westmead Institute of Medical Research, Faculty of Medicine and Health, University of Sydney, Westmead, NSW 2145, Australia

**Keywords:** functional neurological (conversion) disorder (FND), immunisation stress–related responses (ISRRs), dissociative neurological symptom reactions (DNSRs), functional (dissociative) seizures, children, adolescents, young people, SARS-CoV-2 vaccines, COVID-19 vaccines

## Abstract

Background: The emergence of acute-onset functional neurological symptoms, the focus of this study, is one of three stress responses related to immunisation. This case–control study documents the experience of 61 young people with past or current functional neurological disorder (FND) in relation to the COVID-19 vaccination program in Australia. Methods: Information about the young person’s/parent’s choice and response pertaining to COVID-19 vaccination was collected as part of routine clinical care or FND research program follow-up. Results: 61 young people treated for FND (47 females, mean age = 16.22 years) and 46 healthy controls (34 females, mean age = 16.37 years) were included in the study. Vaccination rates were high: 58/61 (95.1%) in the FND group and 45/46 (97.8%) in the control group. In the FND group, 2 young people (2/61, 3.3%) presented with new-onset FND following COVID-19 vaccination; two young people with resolved FND reported an FND relapse (2/36, 5.56%); and two young people with unresolved FND (2/20, 10.0%) reported an FND exacerbation. In the control group no FND symptoms were reported. Conclusions: Acute-onset FND symptoms following COVID-19 vaccination are uncommon in the general population. In young people prone to FND, COVID-19 vaccination can sometimes trigger new-onset FND, FND relapse, or FND exacerbation.

## 1. Introduction

Functional neurological (conversion) disorder (FND) is a neuropsychiatric disorder that involves aberrant changes within and across neural networks [1,2]. Young people with FND present with motor, sensory, cognitive, and seizure symptoms unexplained by other neurological disorders. While the *Diagnostic and Statistical Manual of Mental Disorders* (fifth edition; DSM-5) no longer requires identification of a precipitating stressor, young people with FND—and their families—commonly report a triggering stressor or event against the background of cumulative stress or adverse childhood experiences (ACEs) [3,4]. The types of reported stressors are very broad. Common psychological stressors include worry, loss, and distress pertaining to family, friendship, school, and maltreatment. Common physical stressors include illness (e.g., viral illness), injury (e.g., a fall, sprains, fractures, or a hit on the head), and medical procedures (e.g., surgery, imaging procedures, and vaccinations). Contemporary aetiological models adopt a biopsychosocial network (systems) perspective [1,5,6,7]. These models propose that FND involves complex interactions between biological susceptibility and lived experience, including the effect of that experience on activation of the stress system(s), on the brain as active predictor [8,9,10], on the perception of that experience, and on the biological (epigenetic) embedding of lived experience in the body and brain [2,3,8,11]. In the current study we document, for a cohort of young people with past or current treatment for FND, reports of new-onset, exacerbated, or recurrent functional neurological symptoms following vaccination for COVID-19. In the discussion we apply our current understanding of FND to consider the potential mechanisms by which vaccination may trigger FND symptoms.

At the time of writing, in response to the COVID-19 pandemic, “67.7% of the world population has received at least one dose of a COVID-19 vaccine. 12.58 billion doses have been administered globally, and 4.61 million are now administered each day” [12]. In Australia, the vaccine became available to young people under 16 years of age in early 2021 (see Appendix A for information about COVID-19 rollout in Australia) [13,14,15]. Data pertaining to young people’s adverse responses to the COVID-19 vaccine is just beginning to emerge.

Vaccine hesitancy pertaining to the COVID-19 vaccine has been documented among people with chronic neurological disorders, including those that are functional [16,17]. In our own clinical setting, many parents of young people with FND have expressed anxiety about potential neurological complications of the COVID-19 vaccine—and other vaccina-tions—with particular worries pertaining to relapse or worsening of the young person’s FND symptoms. Many parents have requested our clinical opinion with regard to the po-tential risks of vaccination. In our responses to these parents, we have acknowledged both the difficulty of the decision-making process and the need to consider the risks to their child (see section below). We have communicated that from our medical standpoint the risks of not being vaccinated outweigh the risks of vaccination. An unvaccinated young person has an increased risk of experiencing a more severe illness and of developing long COVID, with gaps in knowledge about long-term outcome [18,19,20,21,22,23,24]. We have also pointed the family to available information resources [25]. During this conversation with the par-ents, we mention that all vaccinations, and not just the ones for COVID-19, are associated with a small risk of complications (see below) and that, while the risk of complications from COVID-19 vaccination is small, it is not yet possible to identify the individuals who might be affected.

Existing literature suggests that neurological complications following COVID-19 vaccination are rare and occur in a small subset of cases [26,27,28]. Some complications are understood to reflect a response to the vaccine constituents [27], whereas others reflect a range of stress-related responses—termed *immunization stress–related responses* (ISRRs) [29]. A historical and cross-cultural analysis suggests that ISRRs “have been observed in different cultures, particularly in children and adolescents” [30] (p. 330). Stress-related re-sponses can be divided into three clusters:*Acute anxiety-related responses* that manifest just prior to, during, or after the administration of the vaccine (e.g., fainting [vasovagal response], palpitations, hyperventilation, and fear cognitions) [29].*Acute-onset functional neurological symptoms* (±nonspecific functional symptoms) that manifest hours or days after the vaccination (e.g., weakness or paralysis, shaking, twitching and abnormal movements, limb posturing, gait irregularities, speech difficulties, and functional seizures) [29,31,32,33]. Using European terminology, the World Health Organization calls acute-onset functional neurological symptoms dissociative neurological symptom reactions (DNSRs) [29]. We retain the terms functional neurological symptoms and functional neurological disorder, which are used by the large majority of researchers in the field.*A post-immunisation illness* characterised by nonspecific functional somatic symptoms such as headache, dizziness, nausea, dyspnoea, fatigue, and generalised sense of weakness [27,33].

The mechanisms underpinning the first cluster, acute anxiety-related responses, are well understood. Fainting (vasovagal response) and palpitations involve sudden activa-tion of the defensive vagal and sympathetic components of the autonomic nervous sys-tem, respectively (for review, see Chapter 6 in Kozlowska et al. (2020) [7]). Stress-related hyperventilation involves activation of the autonomic nervous system coupled with acti-vation of the respiratory motor system (for review, see Chapter 7 in Kozlowska et al. (2020)). Hyperventilation and low pCO_2_ are associated with a cascade of neurophysio-logical changes that underpin a broad array of functional somatic symptoms (see Appendix A) [34,35]. Fear cognitions are known to arise in neurophysiological states of high arousal triggered by fear stimuli, such as a medical procedure (including vaccination) [36].

By contrast, the mechanisms underpinning the second and third clusters—functional neurological symptoms and post-immunisation illness—are less well understood, poten-tially much more complex, and the focus of current research efforts (see discussion with regard to functional neurological symptoms).

In this article we document the experience of 61 young people with past or current treatment for FND in the Mind–Body Program (2018–2022) in relation to the New South Wales COVID-19 vaccination program and the occurrence—new onset, exacerbation, or recurrence—of acute-onset functional neurological symptoms (cluster 2 above). We also report the experience of healthy controls who volunteered to take part in the FND research program.

## 2. Methods

### 2.1. Participants

Participants were 61 young people who had been referred for treatment for FND in the Mind–Body Program at The Children’s Hospital at Westmead (October 2018 to September 2022) and who had agreed to participate in the FND research program. All young people with FND had undergone a comprehensive neurology assessment and had been given a positive diagnosis of FND (DSM-5 criteria) by a paediatric neurologist (see Appendix A for positive rule-in signs for FND) [37,38]. Four participants who had participated in other components of the research program could not be contacted, and one parent decided not to participate in the current component. Forty-six healthy controls had been recruited from the same age bracket and geographical catchment area. Control participants were screened for the absence of mental health disorders, history of head in-jury, family history of mental health disorders, and chronic health concerns. Control par-ticipants constituted a control group for previous study components looking at biological markers [2,3], and they likewise constituted a healthy control group for the current study.

On admission to the research program, 60/61 young people with FND and all healthy controls completed the Early Life Stress Questionnaire (ELSQ) (see Table 1) and the Depression Anxiety and Stress Scales (DASS-21) (see Appendix A for description of measures) [39,40,41]. One young person with FND had been too ill to complete the measures on initial presentation. All young people were rated on the Global Assessment of Functioning scale (GAF).

The Sydney Children’s Hospital Network Ethics Committee approved the study (HREC/18/SCHN/232). Participants and their legal guardians provided written informed consent.

### 2.2. Data Acquisition

In August/September 2022, information about the young person’s/parent choice per-taining to COVID-19 vaccination was collected as part of routine clinical care or as part of follow-up—looking at outcomes over time—for the FND research program. For young people treated in the past, questions were asked over the phone (see Appendix A for script used to guide the phone interviews). For young people presenting for treatment during this study, the same questions were asked face-to-face. If the young per-son had been vaccinated, the young person and parent were asked about the number of doses the young person had received. They were also asked, using an open-ended ques-tion, whether the young person had experienced any issues with the COVID-19 vaccina-tion. The interviewers (NL and KK) did not offer any suggestions as to what “possible is-sues” could entail. For the small subset of young people who presented with FND follow-ing COVID-19 presentation, the information pertaining to the young person’s response to vaccination was part of the clinical history on presentation.

*Acute-onset FND* was defined as new FND symptoms in a young person who had never experienced FND symptoms before. *FND relapse* was defined as relapse of FND symptoms in a young person who had previously been diagnosed with FND and who had fully recovered from all FND symptoms prior to vaccination. *FND exacerbation* was defined as an increase in intensity or number of FND symptoms in a young person with current FND.

### 2.3. Analysis of Clinical Characteristics and Self-Report Data

Using SPSS Statistics 26, we performed Chi-square analyses and independent t-tests to calculate differences between the FND and control groups on, respectively, categorical and continuous variables.

## 3. Results

### 3.1. Participant Characteristics

The final study group interviewed about their vaccination experience (August/September 2022) comprised 61 young people (47 girls and 14 boys) aged 12.08 to 21.83 years (mean = 16.22, SD = 2.26; median = 15.92) who had been or were being treated for FND and 46 healthy controls (34 girls and 12 boys) aged 10.83 to 21.5 years (mean = 16.37, SD = 2.82; median = 16.04). The groups were matched for sex (χ^2^ = 0.14; *p* = 0.708) and age (t(84.40) = −0.303; *p* = 0.763).

In the FND group, the clinical presentations of the 61 young people—at the time of their admission for treatment into the Mind–Body Program—had been diverse (see Figure 1). They presented with one or more functional neurological symptoms (range, 1–9; mean = 3.32, SD = 1.96; median = 3.00) and significant functional disability (GAF scores ranging from 10 to 51 (mean = 31.77, SD 9.26; median = 31.00) (see Appendix A). On presentation for treatment, 53 (86.9%) had suffered from a comorbid mental health disorder. Anxiety (*n* = 51, 83.6%) and depression (*n* = 31, 50.8%) were the most common). In addition, 21 (34.4%) had reported suicidal ideation, and 16 (26.2%) had suffered from one or more comorbid functional syndromes: functional gut disorder (*n* = 10, 16.4%), postural orthostatic tachycardia syndrome (POTS) (*n* = 6, 9.8%), and frequent vasovagal events (*n* = 1, 1.6%).

### 3.2. Self-Report Measures and Heart Rate on Admission to the Research Program

On admission to the research program, young people in the FND group reported a greater number of ACEs (total ELSQ score: range, 0–10; mean = 3.58, SD = 2.80; median = 2.00) compared to healthy controls (range, 0–3; mean 0.52, SD = 0.78; median = 0.61) (t(70.57) = 8.22; *p* < 0.001)) (see Table 1). On admission to the research program, young people in the FND group also reported higher levels of distress (total DASS score: range, 3–52; mean = 27.68, SD = 12.86; median = 28) compared to healthy controls (range, 0–30; mean = 5.72, SD = 5.67; median = 4.00) (t(85.50) = 11.82; *p* < 0.001)).

### 3.3. Health Status at Time of Vaccination (or Option of Vaccination)

Two young people in the FND group (2/61, 3.3%) were referred for treatment to the Mind–Body Program after their FND illness was triggered by COVID-19 vaccination. For the rest of the cohort (*n* = 59), their medical status at the time of vaccination—or when pre-sented with the option of vaccination—was as follows: fully resolved FND (*n* = 31, 50.8%); resolved FND with a pattern of short-lived relapses in the face of stress (*n* = 7, 11.5%); and unresolved FND (*n* = 20, 32.8%) (see Table 2).

Forty-seven (77.0%) young people in the FND group were experiencing mental health symptoms at the time of vaccination (or option of vaccination), with anxiety (*n* = 34, 55.7%), depression (20, 32.8%), and PTSD (*n* = 5, 8.2%) being the most common (see Table 2). Eight (13.1%) young people in the FND group suffered from an ongoing functional disorder (other than FND) at the time of vaccination (or option of vaccination) (see Table 2). One suffered from complex chronic pain (see Table 2).

The young people in the healthy control group were all healthy at the time of vac-cination and had not developed any new health concerns since admission into the re-search program.

### 3.4. Vaccination Rates in Young People with FND and Healthy Controls

The vaccination rate in the FND group was 58/61 (95.1%). The vaccination rate in healthy control group was 45/46 (97.8%). There was no statistical difference between groups on vaccination rate (χ^2^ = 0.55; *p* = 0.459).

In the control group, the one unvaccinated adolescent girl (aged 15.4 years)—and her family—reported that she had not had the vaccine because her mother had experienced neurological symptoms (loss of words, saying the wrong word, forgetting how to spell things) and cardiac symptoms (difficulties breathing, arrythmia, and chest pressure) fol-lowing COVID-19 vaccination. In this context the adolescent (and her parents) were con-cerned that the vaccine could potentially trigger cardiac and neurological symptoms in the adolescent.

### 3.5. Rates of FND Symptoms in Response to the COVID-19 Vaccine as Reported by the Young Person and Family

The rates of FND relapse, FND exacerbation, and first-presentation FND following COVID-19 vaccination in our FND cohort are reported in Figure 2. Below we provide case vignettes of the six individuals who reported adverse FND responses to the Pfizer vac-cine—the only vaccine available for their age group at the time they were vaccinated. We present the cases in the order of their admission into the Mind–Body Program. In cases 2, 5, and 6—one relapse and two new-onset FND—the FND diagnosis was confirmed by a neurologist at our hospital. In cases 1, 3, and 4 the diagnosis was confirmed by the family doctor, who organised further physiotherapy and mental health treatment with physio-therapists and mental health clinicians. The vaccination record of all six cases is reported in Table 3 and the medication history (during the FND intervention) is reported in Appendix A.


**Case 1. Relapse of FND symptoms post COVID-19 vaccination in a child whose FND had resolved**


An 18-year-old female with a past history of FND (onset of functional seizures, leg weakness and uncoordinated gait, and blurred vision, all at age 15), coupled with comor-bid back pain, nausea, fatigue, dizziness, and functional gut symptoms, presented with FND relapse. Three years earlier, the initial episode of FND was triggered by a sporting injury (a fall causing a fracture). Other stressors at the time had included bullying and the death of her grandfather from a long illness. The young woman had recovered from the FND illness more than two years earlier. At the time of vaccination, she had current diag-noses of POTS and anxiety. Two weeks after her first COVID-19 Pfizer vaccination, she developed recurrence of functional seizures, which continued every 2–3 days for six months before resolving with further treatment. She subsequently had two further vac-cinations (Pfizer and Moderna) with no issues (see Table 3).


**Case 2. Relapse of FND symptoms post COVID-19 vaccination in a young woman whose FND had resolved but who had experienced a number of short relapses with stress**


A 17-year-old female with developmental disability and severe anxiety developed her first episode of FND at age 15. That episode had emerged in the following context: influenza vaccination; chest infection (that same week); allergic reaction (after eating an egg sandwich) with urticarial rash and difficulty breathing (week 3); extreme fatigue and a collapse event at school, with hypokalaemia on blood screen treated with potassium (week 4); and emergence of FND in the form of jerky movements (week 5). Over time, the patient’s FND symptoms came to include back arching (dystonic movements in her back), functional seizures, loss of coordination in the legs (astasia abasia), and shaking and tremoring, coupled with episodes of sudden stabbing pain and panic attacks (with long periods of hyperventilation). The death of a grandfather was an additional, concurrent stressor. Following full recovery, the patient had had short-lived FND relapses in the con-text of stress at school (leg weakness/paralysis, difficulties swallowing, intermittent loss of vision, loss of control of her bladder, and functional seizures). She presented two days after her first COVID-19 Pfizer vaccination with dysphagia, sialorrhea, leg weakness, and exacerbation of functional seizures. She was admitted for nine days of rehabilitation as her dysphagia resulted in restriction of oral intake to the degree of becoming medically unstable. Her FND symptoms resolved with treatment. She subsequently had two further COVID-19 Pfizer vaccinations with no issues (see Table 3).


**Case 3. Worsening of FND symptoms in unresolved FND**


A 19-year-old birth-assigned female who identifies as male suffered with chronic FND from age 15: functional seizures, leg weakness (initially requiring a wheelchair but subsequently resolved), and time-limited episodes of whole-body stiffening and functional blindness that occurred on a weekly basis. The patient’s FND had occurred in the setting of cumulative stressors over many years. These included: bullying, exposure to violence, and exposure to a severe mental illness in an older sibling. Comorbid diagnoses at the time of vaccination included dissociative identity disorder, depression, and gender dysphoria. Three days after the second COVID-19 Pfizer vaccination, the patient experienced a recurrence of bilateral leg weakness and required a wheelchair to mobilise. The leg weakness continued for six months while the patient underwent outpatient rehabilitation. The time-limited FND episodes of whole-body stiffening and blindness continued to occur. She subsequently had a further COVID-19 Moderna vaccination with no issues (see Table 3).


**Case 4. Worsening of FND symptoms in unresolved FND**


A 15-year-old adolescent female aspiring to be a doctor first experienced FND symp-toms at age 13 while under academic stress. Her FND included weakness and dragging of the left leg, loss of coordination in the legs, tremor, loss of hearing and tinnitus, blotchy vision, light-headedness, nausea, and cognitive symptoms of being unable to think clearly. In the patient’s own words, “The root cause of my symptoms was my tendency to put pressure on myself. I just challenged myself so much that I eventually burnt out.” At the time of her first COVID-19 Pfizer vaccination, the patient was making excellent progress toward a full recovery. Her remaining symptoms included functional seizures (of decreasing frequency), tinnitus, and visual loss, as well as comorbid POTS and functional gut symptoms. Immediately following the first vaccination, the patient experienced a functional seizure. Following the functional seizure, her legs were weak and unsteady, she experienced ticcing a new symptom (functional tics), and she needed to use a wheelchair. The leg weakness recurred following the second vaccination. On both occasions the symptoms resolved in a few days.


**Case 5. New-onset FND following COVID**
**-19 vaccination**


A 17-year-old female experienced new-onset FND following COVID-19 vaccination that led to admission into the Mind–Body Program for treatment. Immediately after her first Pfizer vaccination, she experienced pain and itchiness at the injection site. She woke the next day with a circular, raised, erythematous rash with central blistering that ex-tended along the entire right arm. The rash remained for a week. The day following the vaccination she also experienced nausea and vomiting as well as an unresponsive epi-sode that lasted for approximately 40 min.

Three weeks following the vaccination, she had increased frequent collapses. She was admitted for further investigation: an electroencephalogram did not show any evidence of epileptiform activity. Her collapses continued for six months before the diagnosis of FND was made by a neurologist after further investigations (lumber puncture and magnetic resonance imaging). By this time the patient was experiencing new symptoms of limb weakness, muscle spasms (causing a shoulder subluxation), and functional tics, which were both verbal and motor. She was also experiencing irritable bladder, functional gut symptoms, and dizziness on standing secondary to POTS.

Predisposing risk factors included the following: the death of four grandparents, death of an aunt, death of a close friend, death of three dogs, and sexual assault, all within the period of two years. Prior to the development of FND, this patient had been given a diagnosis of depression with posttraumatic stress disorder—related her traumatic expe-riences—and she was being managed by a community mental health team.

The patient was admitted for a three-week rehabilitation admission. On discharge she was mobilising on a walking frame, and her tics had largely subsided. She had en-gaged in the tasks of learning physiological regulation strategies [7] and learning to ad-dress illness-promoting cognitive processes [42]—rumination, catastrophising, and so on—that functioned to maintain activation of her stress system. A graded return to school was organised. For the longer term, the plan was for the patient to engage in trauma-focused therapy over a 6–12 month period.


**Case 6. New-onset FND following COVID-19 vaccination**


A 12-year-old girl experienced new onset of three different medical conditions—atypical narcolepsy (type 2, HLA negative) [43,44,45,46,47,48,49,50,51,52,53] (see Appendix A), a functional gut disorder [53], and FND (see Appendix A)—following her second Pfizer vaccination. The narcolepsy diagnosis was made by a sleep physician (KW) based on clinical presentation (persisting pattern of daytime sleepiness), polysomnog-raphy (short REM latency and sleep fragmentation), and a multiple sleep latency test (falling asleep on all four nap opportunities coupled with three sleep-onset REM periods). Measurements of cerebral spinal fluid orexin (hypocretin) were not available. The functional abdominal pain diagnosis was confirmed by a gastroenterologist. The FND diagnoses—persistent motor weakness of the lower limbs (unrelated to sleep) and functional/dissociative seizures—were confirmed by a neurologist via rule-in positive neurological signs and video EEG (gold-standard assessments) [38,54,55] (see Appendix A).

The child presented to the emergency department with an 11-day history of hyper-somnolence that started two hours after her second vaccination. She was treated with dexamphetamine. Four weeks following the vaccination, she developed acute episodes of nausea, vomiting, and abdominal pain, coupled with an inability to tolerate oral intake. The patient’s pain was managed with simple analgesics; her distress, with olanzapine (as needed); and her nutritional needs, with feeds via a nasogastric tube. Two weeks follow-ing the vaccination, she developed bilateral lower limb weakness (necessitating a wheel-chair to mobilise). She also developed functional seizures that showed semiology changes across time: zoning out; episodes of collapse; generalised shaking; generalised shaking coupled with the patient hitting herself; episodes of aggression while the patient was in an altered state; and episodes where the patient ran around and around her hospital room while in an altered state (although she was otherwise still unable to use her legs when in her usual state of mind). The child’s FND symptoms made the clinical picture very complex and excluded the possibility of clearly diagnosing cataplexy.

The patient was discharged from hospital on day 97 with ongoing outpatient man-agement of narcolepsy (by the sleep team), functional gut disorder (by the family doctor), and FND, depression, and anxiety (by the local mental health team).

Predisposing risk factors for FND and a functional gut disorder included the follow-ing: the stress of home schooling during COVID-19 lockdowns; increased stress (conflict) in the parental relationship, which had previously been amicable; conflict with peers in Year 6; loss of the opportunity to go to a performing arts high school; stress around transi-tion to high school; and a medical history of functional cough, functional abdominal pain, and untreated depression.

### 3.6. Rate of COVID-19 Infection in Young People with FND and Healthy Controls

In the FND group, 24 (39.3%%) reported that they had contracted COVID-19. Of these, 23 (37.7%) reported that they had contracted COVID following one or more vaccinations. No exacerbations of FND were reported in association with COVID-19 infections, and no prolonged illness (long COVID) was reported.

In the control group, 31 (67.4%) reported that they had contracted COVID-19. Of these, 30 (96.8%) reported that they had been vaccinated, and they had contracted COVID-19 after one or more vaccinations. In this vaccinated subgroup, no neurological symptoms and no prolonged illness (long COVID) were reported.

Additionally, in the control group, the adolescent girl who had not been vaccinated—because her mother had experienced neurological symptoms after being vaccinated—reported a prolonged illness (long COVID). After falling ill with COVID-19, she developed POTS—reflecting activation and dysregulation of the autonomic nervous system—alongside high temperatures and cold-like symptoms. The POTS symptoms slowly resolved after more than two months and recurred three times with subsequent viral infections (not COVID-19). In each case the autonomic symptoms preceded the adolescent’s cold-like symptoms. The symptoms were severe enough to require management by a cardiologist.

## 4. Discussion

The current study documents the experience of a cohort of 61 young people admitted to the Mind–Body Program for the treatment of FND (October 2018 to September 2022)—and 46 healthy controls—with respect to the COVID-19 vaccination program in New South Wales, Australia. Vaccination rates across the two groups were high. In the FND group (see Figure 2), two young people (2/61, 3.38%) developed new-onset FND following COVID-19 vaccination; two young people with resolved FND experienced an FND relapse (2/36, 5.56%) following vaccination; and two young people with unresolved FND (2/20, 10.0%) experienced FND symptom exacerbation following vaccination. In the healthy control group (see Figure 2), no young people experienced new-onset FND fol-lowing COVID-19 vaccination. However, one control opted out of the vaccination program because her mother had experienced post-vaccination neurological and cardiac symptoms. This unvaccinated control case experienced long COVID after contracting COVID-19 infection. Our study findings cohere with the broader literature. While functional neurological complications following COVID-19 vaccination are rare, they do occur in a small subset of susceptible young people. Data from this small cohort suggest that in susceptible individuals, COVID-19 vaccination can trigger new-onset FND, relapse in resolved FND, or exacerbation of FND symptoms in unresolved FND.

Three previous follow-up studies of young people with FND at 4 years, 12 months, and 18 months have shown that 17.5%, 11.5%, and 4%, respectively, experienced relapse in response to subsequent stress. In the current study, of the 51 young people with past or current FND who were vaccinated for COVID-19, 4/51 (7.84%) experienced a relapse or a worsening of their symptoms with vaccination (see Figure 2). As above, the findings sug-gest that for a small subset of young people prone to FND, vaccination can function as an illness trigger.

Stress-related responses to vaccination are well documented in the literature (see the three clusters described in the introduction) [27,29,30,31,32,33]. As mentioned in the introduction, other studies have reported three types of stress-related responses to vaccination: acute anxiety-related responses (cluster 1), acute-onset functional neurological symptoms (clus-ter 2), and post-immunisation illness characterised by nonspecific functional symptoms (cluster 3). An unexpected finding in the current study is that none of the participants re-ported any stress-related symptoms under clusters 1 or 2. It is possible that our open-ended question—”Did you experience any issues with the COVID-19 vaccination?”—which did not include any suggestions about what we may have been looking for, yielded a lower report rate than questions that specifically asked about the experience of particular post-vaccination symptoms. We intentionally opted to use the open-question methodology because studies with adults with FND suggest heightened suggestibility—that is, heightened responsiveness to direct verbal suggestions [56]. Heightened suggestibility is also apparent in clinical work with young people with FND—so much so that careful use of language that encourages health-promoting expectations, not illness-promoting ones, is a key element of clinical practice [7,57]. In this context we preferred to err on the side of underreporting than that of overreporting (potentially as a function of suggestibility).

In the clinical setting, the understanding of underlying biological mechanisms that underpin stress-related responses to vaccination is limited. For example, the 2019 updated World Health Organization user manual—Causality Assessment of an Adverse Event Following Immunization [29]—FND presentations triggered by vaccination are described as having “no apparent physiological basis” (p. 48). This framing of the FND as a negative diagnosis—as a diagnosis of exclusion having no apparent physiological basis—does not meet current best practice guidelines. Guidelines highlight the importance of providing the patient with a positive diagnosis of FND (see Appendix A) together with an explanation of the diagnosis [58]. In the section that follows we provide a short summary of current thinking about FND, which we hope will enable clinicians to provide patients with an explanation based on current research. Patients typically receive these explanations with gratitude, even when the clinician highlights that the explanation re-flects our best current *hypothesis* based on emerging research.

A large body of research highlights that stressors—whether they be physical or psy-chological—activate the stress response in a coordinated manner (see Figure 3) [59]. By the same token, the process of being vaccination can be conceptualised to function as a physical (bottom-up) stressor or a psychological (top-down) stressor.

Vaccination as a physical (bottom-up) stressor involves activation of the immune-inflammatory system via vaccine constituents and the mounting of an antibody response. The antibody response also involves activation of the sympathetic nervous system (a component of the autonomic nervous system) and the HPA axis [61]. Because the sympathetic system and HPA axis work as coupled systems, they are sometimes referred as the sympathetic-adreno-medullary (SAM) axis. When the stress response is activated, the SAM axis secretes epinephrine (adrenalin) and norepinephrine (noradrenalin), and the HPA axis secretes glucocorticoids. In this way, the vaccination as physical stressor activates multiple components of the stress system “bottom-up”.

Vaccination as a psychological (top-down) stressor involves activation of the stress system via mental processes (thoughts and feelings) [59,62,63,64]. Some young people may experience fear, anxiety, negative beliefs, and negative expectations pertaining to the pro-cess of vaccination—having a needle. Other young people may hold beliefs about poten-tial negative outcomes of vaccination on health and wellbeing. These beliefs may reflect those held in the family, expectations communicated by media, or knowledge about an adverse outcome in a family member or friend. In this way the vaccination as a psycho-logical stressor activates multiple components of the stress system “top-down”.

Previous literature has noted that young people are susceptible to communications of threat (and potential threat) from family members, peers, and social media. A group-contagion effect pertaining to mass outbreaks of FND and other functional somatic symptoms (see clusters 2 and 3 in the Introduction) is well documented. Such outbreaks appear to occur during times of threat or stress, including war and pandemics [65,66]. The same phenomenon—clusters of functional illnesses—has been reported in relation to vaccination campaigns [33,67,68]. For example, following the H1N1 vaccination in Taiwan and South Korea, school-based vaccination was associated with a higher proportion of cases (functional illness) than individual vaccination [33]. Yang highlights that “a chain reaction following the index case may attract mass media attention and is promptly spread by the people who share similar beliefs about the vaccine and its safety” [33] (p. 31). In this way “top-down” cognitive factors—in this case, beliefs—coupled with stress-system activation (and activation of the motor respiratory system), can set the stage for triggering an FND illness or other functional somatic symptoms (see Appendix A and Figure 3).

The hypothesised mechanisms—working bottom-up and working top down—are not mutually exclusive. The mechanisms can occur in tandem and may reflect a set of neuroimmune–endocrine–psychological cascade of threat-related responses—the body–brain–mind response to signals of danger (physical or psychological)—that interact with each other in a mutually reinforcing way.

Importantly, in most healthy young people, stress-system activation is short-lived. The stressor is dealt with, and body and brain systems return to normal function (that is, to a level of function within normal homeostatic parameters). In young people who de-velop FND, however, various components of the stress system—the HPA axis [3], the au-tonomic nervous system [69,70,71], the immune-inflammatory system [72], and subjective distress [2]—remain activated or dysregulated over time. Studies suggest that in young people with FND, the flow-on effect of this cascade of stress-related responses is sustained activation of neural (neuron–glial) networks [73,74,75] coupled with an alteration of connec-tivity within and between networks [2,11]. These aberrant changes within and across networks are thought to underpin the motor, sensory, and cognitive symptoms that typify the illness [1,2].

In this final section of the discussion we consider three potential biological mecha-nisms via which vaccination may trigger a cascade of stress-related responses and con-tribute to the process by which FND symptoms are triggered or exacerbated.

First, the nervous system and the immune system are interdependent systems that communicate in a bidirectional manner—from the brain to the immune system and from the immune system to the brain [61]. Any alteration in one of the systems—for example, immunological stress such as vaccination—has the potential to affect the functioning of the other system via immune-neuro and neuro-immune cross-talk [61]. The endocrine system also takes part in this cross-talk. Important neuroimmune mechanisms—presumably also relevant to acute-onset FND—are summarised in Appendix A [61,76,77,78,79,80] (for review, see Dantzer, 2018).

Second, neural networks are now conceptualised as neuron–glia networks—where microglia engage in reciprocal signalling with neurons to underpin brain function in health and disease [81,82,83,84,85]. Microglia are the resident macrophage population of the cen-tral nervous system [86]. Because microglia activate both in response to physical (includ-ing immunological) and psychological stress, they play an important role in sculping the brain in the context of early life adversity [76]. Consequently, glial cells are hypothesised to play an important role in stress-related disorders. They are thought to function as neu-roimmune sensors of stress [77,79,87] and are hypothesised to play a role triggering and maintaining aberrant neural network function in FND and other stress-related disorders [74,88].

Third, young people with FND [4,89] report an increased number of adverse child-hood experiences [4,89]. Cumulative ACEs are associated with repeated stress-system ac-tivation [60,90,91,92]. Recurrent HPA-axis activation supports epigenetic reprogramming that prepares the young person’s biological systems to mount a robust—and sometimes excessive—stress response in the face of future stress. At the molecular level, the effects of HPA-axis activation and glucocorticoid signalling are largely mediated by the glucocorticoid receptor that drives the genomic actions of glucocorticoids, mediating the biological embedding of experience via plasticity changes in every tissue of the body (including the brain) [60,91]. Along these lines, a recent study with adults showed that aberrant changes in neural network function were influenced by the severity of early life physical abuse and that the connectivity maps that were correlated with physical abuse overlapped with differences in gene expression in three gene clusters [11].

Once vaccination or another stressor (physical or emotional) has triggered the FND symptoms, a number of factors may contribute to symptom maintenance (see Appendix A) [3,6,7,8,9,10,11,36,42,69,70,71,72,73,74,75,88,93,94,95,96]. Broadly speaking, all these factors re-flect the young person’s inability, including the inability of the young person’s biological system, to activate restorative (calming) processes that could help the mind, body, and brain return to baseline function. In this context, a range of threat-related processes—both psychological and neurophysiological—continue unabated and become the new norm for that person.

## 5. Limitations

A prospective design using a whole-population approach is needed to provide inci-dence statistics of acute-onset FND following COVID-19 vaccination in the general popu-lation. Such data will also tell us what FND symptoms are most common post COVID-19 vaccination. Such a design would also provide incidence data about the two other stress responses related to immunisation: acute anxiety-related responses (cluster 1) and a post-immunisation illness characterised by nonspecific functional somatic symptoms (cluster 3) (see introduction for the description of the three clusters). The lack of longitudinal outcome data for case 5 and case 6—who were still experiencing FND symptoms at the time this paper was written—is another limitation.

## 6. Conclusions

Acute-onset FND symptoms following COVID-19 vaccination are uncommon in the general population. Notwithstanding, in young people prone to FND, COVID-19 vaccina-tion can trigger new-onset FND, FND relapse, or FND exacerbation. The biological mech-anisms involved are thought to be complex: a cascade of stress-related responses (includ-ing epigenetic reprograming) that lead to a sustained activation of neural (neuron–glial) networks [73,74,75] coupled with an alteration of connectivity within and between networks [2,11]. Vaccination appears to act as a physical or psychological stressor that triggers the stress system and its cascade of threat-related responses, with a final endpoint of aberrant neural network function that supports new-onset functional neurological symptoms. Akin to FND triggered via ACEs, FND triggered by vaccination requires prompt diagnosis and treatment to maximise the likelihood that the young person will return to health and wellbeing [97]. Presence of multiple comorbidities—medical or psychiatric—may complicate the treatment process and adversely affect long-term outcomes.

## Figures and Tables

**Figure 1 vaccines-10-02031-f001:**
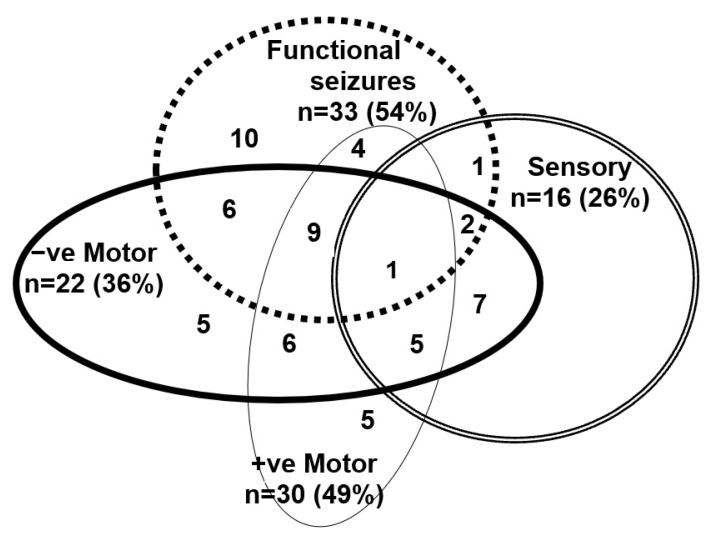
Visual depiction of functional neurological symptoms—past of current—experienced by the young people in the FND cohort.

**Figure 2 vaccines-10-02031-f002:**
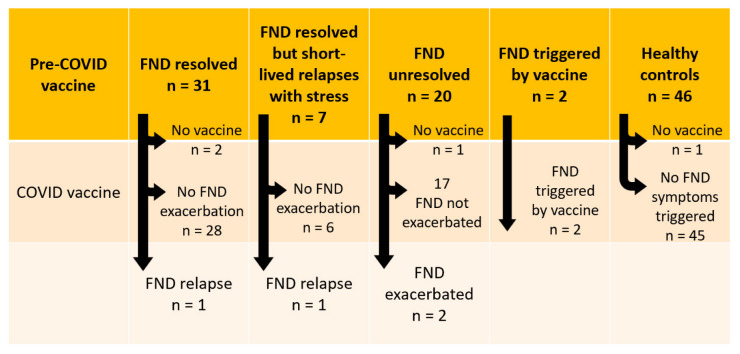
Visual representation of vaccination responses.

**Figure 3 vaccines-10-02031-f003:**
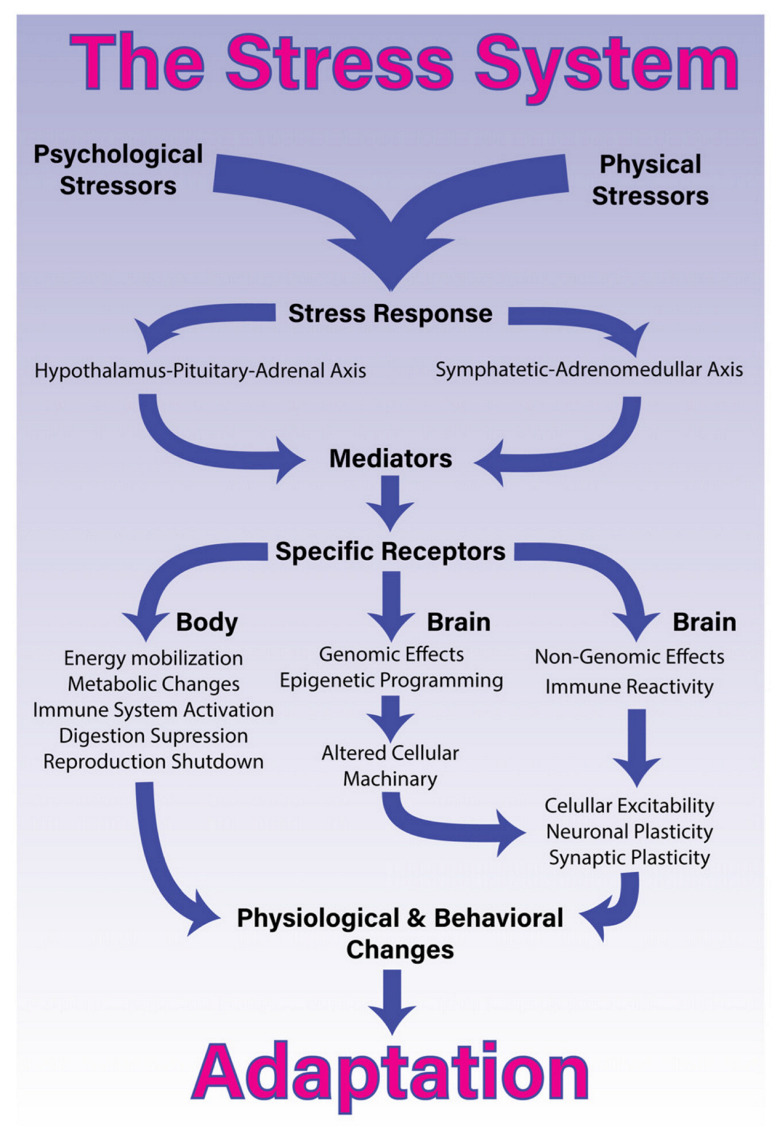
The stress system. “The stress system. Processing and coping with stressful situations requires the engagement of complex mechanisms that integrate brain and body. The response to stressful stimuli is articulated by a wide diversity of brain structures that collectively are able to detect or interpret events as either real or potential threats (stressors). The perception of these events as stressors involves different networks depending on whether it is a physical or psychological stressor. The identification of a stressor leads to activation of two major constituents of the stress system and the release of its final mediating molecules. The sympathetic-adreno-medullary (SAM) axis, secretes epinephrine [adrenalin] and norepinephrine [noradrenalin] and the hypothalamus-pituitary-adrenal (HPA) axis, secretes glucocorticoids. Once these axes are activated in response to a given stressor, they will generate a coordinated response that starts within seconds and might last for days, providing quick responses enabling both an appropriated strategy almost immediately, and homeostasis restoration. To accomplish this, the stress response systemically promotes energy mobilization, metabolic changes, activation of the immune system, and suppression of the digestive and reproductive systems. More specifically in the brain, the stress response induces short- and long-term effects through non-genomic, genomic and epigenetic mechanisms. These central effects, combined with pro-inflammatory signalling, lead to alterations in cellular excitability as well as synaptic and neuronal plasticity. Collectively, these body-brain effects mediate alterations in physiology and behaviour that enable adaptation and survival” [59] (p. 3). Non-genomic effects of glucocorticoids include, for example, the stimulation of both endocannabinoid production and glutamate release [60]. © 2018 Godoy, Rossignoli, Delfino-Pereira, Garcia-Cairasco and Umeoka [59]. Figure 1 of Godoy LD, Rossignoli MT, Delfino-Pereira P, Garcia-Cairasco N and Umeoka EHL (2018). A Comprehensive Overview on Stress Neurobiology: Basic Concepts and Clinical Implications. Front. Behav. Neurosci. 12:127. https://doi.org/10.3389/fnbeh.2018.00127. Published under Creative Commons Attribution License CC BY (https://creativecommons.org/licenses/by/4.0/).

**Table 1 vaccines-10-02031-t001:** Adverse childhood experiences reported by young people with FND (*n* = 60) * on the ELSQ on admission to the Mind–Body Program for treatment and for participation in the FND Research Program.

Adverse Child Experience	Number Reporting ACEs	Percentage
Bullying/rejection by peers	34	56.7%
Parental separation/divorce	22	36.1%
Family conflict	20	32.8%
Other trauma	19	31.1%
Emotional abuse	17	27.9%
Born prematurely/birth complications	16	26.2%
Domestic violence	13	21.3%
Major surgery/repeated hospitalization	12	19.7%
Separated long period parent/sibling	12	19.7%
Natural disaster first-hand witness	11	18.0%
Physical abuse	10	16.4%
Life-threatening illness parent/sibling	7	11.5%
Sexual abuse	6	9.8%
Extreme poverty or neglect	5	8.2%
Life-threatening illness of injury	4	6.6%
Death of parent/sibling	4	6.6%
House destroyed by fire/other means	3	4.9%
Adopted	0	0%
Witness warfare	0	0%

* One young person with FND was too ill to complete the questionnaires on admission into the research program.

**Table 2 vaccines-10-02031-t002:** Clinical status of young people in the FND group at time of vaccination (or vaccination opportunity).

Group	Number in Group	Comorbid Mental Health Condition	Comorbid Functional Syndrome
FND resolved	31	*n* = 20 Anxiety (*n* = 15) Depression (*n* = 7) PTSD (*n* = 2)	*n* = 4 POTS (*n* = 2) Complex/chronic pain (*n* = 1) Functional abdominal disorder (*n* = 1)
FND resolved but pattern of short-lived relapsed with stress	7	*n* = 6 Anxiety (*n* = 6) Depression (*n* = 1)	*n* = 0
FND unresolved	21	*n* = 19 Anxiety (*n* = 12) Depression (*n* = 10) PTSD (*n* = 2)	*n* = 3 POTS (*n* = 2) Functional abdominal disorder (*n* = 2)
FND triggered by vaccine	2	*n* = 2 Depression (*n* = 2) Anxiety (*n* = 1) PTSD (*n* = 1)	*n* = 1 Functional abdominal disorder (*n* = 1)
Healthy controls	46	*n* = 0	*n* = 0

**Table 3 vaccines-10-02031-t003:** Vaccination history for cases 1–6.

Year and Vaccine Administered															
Case 1	2003 Infanrix HepB PedvaxHIB Poliomyelitis	2004 Infanrix HepB PedvaxHIB Poliomyelitis	2004 Infanrix HepB Poliomyelitis	2004 PedvaxHIB Priorix	2004 Meningitec	2005 Prevenar 7	2005 Prevenar 7	2005 Varilrix	2008 Infanrix IPV Priorix	2016 Gardasil	2016 Boostrix Gardasil	2016 Gardasil Varilrix	2021 Pfizer Comirnaty	2021 Pfizer Comirnaty	2022 Moderna Spikevax
Case 2	2004 Infanrix HepB PedvaxHIB Poliomyelitis	2004 Infanrix HepB PedvaxHIB Poliomyelitis	2005 PedvaxHIB Poliomyelitis	2005 Menjugate PedvaxHIB Prevenar 7 Priorix	2005 Prevenar 7 Varilrix	2008 Infanrix IPV Priorix	2017 Boostrix Gardasil	2017 Gardasil Varilrix	2020 Afluria Quad	2021 Pfizer Comirnaty	2021 Pfizer Comirnaty	2022 Pfizer Comirnaty			
Case 3	2003 Infanrix HepB Oral Polio PedvaxHIB	2004 Infanrix HepB Oral Polio PedvaxHIB	2004 Infanrix HepB Oral Polio	2004 PedvaxHIB	2004 Meningitec	2008 Infanrix IPV Priorix	2016 Boostrix Gardasil	2016 Gardasil	2017 Gardasil	2021 Pfizer Comirnaty	2021 Pfizer Comirnaty	2022 Moderna Spikevax			
Case 4	2007 Infanrix Hexa Prevenar 7	2008 Infanrix Hexa Prevenar 7 Rotarix	2008 Infanrix Hexa Prevenar 7	2008 Hiberix Meningitec Priorix	2009 Varilrix	2010 Panvax	2010 Panvax	2011 Infanrix IPV Priorix	2021 Pfizer Comirnaty	2021 Pfizer Comirnaty					
Case 5	2005 Infanrix HepB PedvaxHIB Poliomyelitis Prevenar 7	2005 Infanrix HepB PedvaxHIB Poliomyelitis Prevenar 7	2005 Infanrix Hexa Prevenar 7	2006 Hiberix Meningitec Priorix	2007 Varilrix	2008 Vaxigrip	2009 Infanrix IPV Priorix	2018 Boostrix Gardasil 9	2019 Gardasil	2021 Nimenrix	2021 Pfizer Comirnaty				
Case 6	2010 Infanrix Hexa Prevenar 7 Rotarix	2010 Infanrix Hexa Prevenar 7 Rotarix	2010 Infanrix Hexa Prevenar 7	2011 Hiberix Meningitec Priorix	2011 Varilrix	2012 Prevenar 13	2013 Infanrix IPV MMR II	2018 FluQuadri	2022 Pfizer Comirnaty	2022 Pfizer Comirnaty					

## Data Availability

The datasets presented in this article are not readily available because ethics to place data in a public repository was not obtained from the children and families who participated in this study. The data may potentially be made available on request to the authors after review of the request by the Sydney Children’s Hospital Network Ethics Committee. Requestors should subject a data analysis plan before requesting the data. Requests to access the datasets should be directed to K.K., kkoz6421@uni.sydney.edu.au.

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
