# Peer review of "COVID-19 Vaccination in Young People with Functional Neurological Disorder: A Case-Control Study"

_vaccines, 2022, doi:10.3390/vaccines10122031_

Round 1

Reviewer 1 Report

The paper described the effect of COVID-19 vaccination on young people with FND. The vaccination increases the adverse effect on young people with FND but not healthy volunteers. Because the just injection of the vaccine triggers the scary feeling to FND symptoms or the contents in the vaccine did trigger the FND symptoms or not, these are not supported in this study because a placebo in the young people with FND was not tested. However, the paper is essential for the research area of FND.

The method needs to include more information about questions for the young people to be easily understood by the reader of the Journal. Moreover, there needs to be a description of how the authors classified the several FND symptoms from answers to the question.   

Moreover, The paper was too long and should be shortened, especially in the case studies.

The weakest point in the manuscript is that the experiential procedures were not shown in detail.  The authors asked questions to the young people, but there was no description of the contents of the question list.  And the authors did not show how to treat and classify the obtained answers set to gain the FND symptoms from the answer results

These points  should be clear in  the manuscript.   1. What is the main question addressed by the research?  above
2. Do you consider the topic original or relevant in the field? Does it
address a specific gap in the field?  relevant 
3. What does it add to the subject area compared with other published
material?   This study is a case controlled study but the control was only set for healthy volunteers (not placebo treatment ).  There are a slight gap compared to the general scientific study.   4. What specific improvements should the authors consider regarding the
methodology? What further controls should be considered?   NO need for further control setting; clarify the experimental procedure.   5. Are the conclusions consistent with the evidence and arguments presented
and do they address the main question posed? I f the obtained results are right, collusion is consistent with evidence   6. Are the references appropriate? Yes 
7. Please include any additional comments on the tables and figures. NO

Author Response

Reviewer #1

Vaccines

The paper described the effect of COVID-19 vaccination on young people with FND. The vaccination increases the adverse effect on young people with FND but not healthy volunteers. Because the just injection of the vaccine triggers the scary feeling to FND symptoms or the contents in the vaccine did trigger the FND symptoms or not, these are not supported in this study because a placebo in the young people with FND was not tested. However, the paper is essential for the research area of FND.

The method needs to include more information about questions for the young people to be easily understood by the reader of the Journal. Moreover, there needs to be a description of how the authors classified the several FND symptoms from answers to the question. 

To respond to Reviewer #1’s question, we have added a text box with the script we used to guide the phone interview with the young person and their parent.

With regard to the second question pertaining to the FND symptoms, the FND symptoms that the young people experienced in relation to the vaccination are described in detail in the cases.

The FND symptoms that the young presented when admitted to the Mind-Body Program are visually depicted in a Venn diagram (see Figure 1). These were all documented on neurological examination using positive rule-in signs (including video EEG for children with functional seizures).

The medical care of all the young people who have been through the Mind-body Program is transferred back to a local treating team typically made up of a family doctor, a physiotherapist, and a psychotherapist. In this context, the young people in Cases 1, 3, and 4 were managed by the treating team in the community (medical reassessment of the FND by the family doctor, and further treatment by a physiotherapist and mental health clinicians). Importantly, all the young people in this group were very familiar with their FND symptoms. Notwithstanding, we have always asked them to see their family doctor for review when a new symptoms arose, and they did.

The young people in cases 2, 5, and 6—one relapse and one new onset post vaccination—had neurology assessments at our hospital and were subsequently treated in the Mind-Body Program.

This information has been added to the manuscript.

Moreover, The paper was too long and should be shortened, especially in the case studies.

We have shortened the two long cases as requested. We have also reorganised the discussion in order to shorten it.

However, Reviewer #3 asked us to add medication history and full vaccination history, so we can shorten only up to a point. 

The weakest point in the manuscript is that the experiential procedures were not shown in detail.  The authors asked questions to the young people, but there was no description of the contents of the question list.  And the authors did not show how to treat and classify the obtained answers set to gain the FND symptoms from the answer results.

As noted above we have added a text box with the script that was used to guide the interview process. See also previous answer regarding the FND symptoms.  

This study is a case controlled study but the control was only set for healthy volunteers (not placebo treatment).  There are a slight gap compared to the general scientific study.  

Yes, Reviewer #1 is correct. This study is a case-control study in a naturalistic setting. It describes the experience of young people with FND (past or current) and their response to the COVID-19 vaccine. The vaccine was given as part of the New South Wales roll-out in Australia. The study documents the young people’s response to the vaccine as this occurred in the naturalistic setting. The study did not utilise placebo vaccines etc.

The idea of using a placebo is an interesting one. There may, however, be a number of problems associated with implementing a placebo design.

First, such a design would put focus-of-attention onto the vaccination process. In patients with FND, focus-of-attention on symptoms amplifies symptoms. A study like the one suggested could therefore bias patients with FND to experience more vaccination symptoms—by setting up negative expectations—than they normally would. In this context, the strength of the current design is its naturalistic design. No focus-of-attention was put onto vaccination. And the questions asked did not offer the respondents any suggestions or any idea that we, the researchers, expected there to be any issues with the vaccine. The young people simply reported what they experienced.

Second, obtaining Ethics approval for the placebo idea raised above would be very difficult in a paediatric setting. Since engagement of young people in the vaccination process is already such a struggle, I presume that obtaining Ethics approval would be very complicated and likely not possible.

Reviewer 2 Report

A very interesting case-control study for COVID-19 vaccination in Functional Neurological Disorder (FND) and a matched control group. The manuscript is well-written and presented. Only a few comments that in my opinion can improve the manuscript.

1. Please spell the abbreviations the first time appear, even in the abstract for example for FND

2.  Would like to see some statistics support, for example, was there a statistical difference in the vaccination rate between the two groups, rather not, however, please support this via a test for proportions or a Chi-square/Fisher exact test.

Author Response

Reviewer #2

A very interesting case-control study for COVID-19 vaccination in Functional Neurological Disorder (FND) and a matched control group. The manuscript is well-written and presented. Only a few comments that in my opinion can improve the manuscript.

  1. Please spell the abbreviations the first time appear, even in the abstract for example for FND

This has been done. Thank you for picking this up.

  1. Would like to see some statistics support, for example, was there a statistical difference in the vaccination rate between the two groups, rather not, however, please support this via a test for proportions or a Chi-square/Fisher exact test.

The requested chi square statistic has been added to the manuscript (χ2 = 0.55; p=.459).

Reviewer 3 Report

This manuscript is too long. Should shorten it.

Abstract: FND should be “Functional neurological (conversion) disorder (FND)” for the first time appearance.

Vaccination history, in terms of all cases, must be clarified.

For the treatment, history of acknowledged medicine and other medicine should be listed.

The authors should cite the following cite and papers and discussed them.

https://www.gov.uk/government/publications/covid-19-vaccination-resources-for-children-and-young-people

https://www.gov.uk/government/publications/universal-vaccination-of-children-and-young-people-aged-12-to-15-years-against-covid-19/universal-vaccination-of-children-and-young-people-aged-12-to-15-years-against-covid-19

https://www.medrxiv.org/content/10.1101/2022.03.22.22272775v1 (https://www.medrxiv.org/content/10.1101/2022.03.22.22272775v1.full.pdf)

•https://www.ons.gov.uk/peoplepopulationandcommunity/healthandsocialcare/causesofdeath/articles/covid19vaccinationandmortalityinyoungpeopleduringthecoronaviruspandemic/2022-03-22

• Piccini, B., Pessina, B., Pezzoli, F., Casalini, E., & Toni, S. (2022). COVID-19 vaccination in adolescents and young adults with type 1 diabetes: Glycemic control and side effects. Pediatric diabetes, 23(4), 469–472. https://doi.org/10.1111/pedi.13326

Ziv, A., Heshin-Bekenstein, M., Haviv, R., Kivity, S., Netzer, D., Yaron, S., Schur, Y., Egert, T., Egert, Y., Sela, Y., Hashkes, P. J., & Uziel, Y. (2022). Effectiveness of the BNT162b2 mRNA COVID-19 Vaccine among Adolescents with Juvenile-onset Inflammatory Rheumatic Diseases. Rheumatology (Oxford, England), keac408. Advance online publication. https://doi.org/10.1093/rheumatology/keac408

•Lawson-Tovey, S., Machado, P. M., Strangfeld, A., Mateus, E., Gossec, L., Carmona, L., Raffeiner, B., Bulina, I., Clemente, D., Zepa, J., Rodrigues, A. M., Mariette, X., Hyrich, K. L., & EULAR COVAX (2022). SARS-CoV-2 vaccine safety in adolescents with inflammatory rheumatic and musculoskeletal diseases and adults with juvenile idiopathic arthritis: data from the EULAR COVAX physician-reported registry. RMD open, 8(2), e002322. https://doi.org/10.1136/rmdopen-2022-002322

Beckley, M., Olson, A. K., & Portman, M. A. (2022). Tolerability of COVID-19 Infection and Messenger RNA Vaccination Among Patients With a History of Kawasaki Disease. JAMA network open, 5(8), e2226236. https://doi.org/10.1001/jamanetworkopen.2022.26236

Author Response

Reviewer #3

This manuscript is too long. Should shorten it.

We have shortened the two long cases, and we have also reorganised the discussion in order to shorten it.

Abstract: FND should be “Functional neurological (conversion) disorder (FND)” for the first time appearance.

Thanks. We have written out FND in the abstract.

Vaccination history, in terms of all cases, must be clarified.

A table detailing the vaccination history for all cases has been added.

For the treatment, history of acknowledged medicine and other medicine should be listed.

FND is not treated with medication. If medications are used, they are used to target comorbid anxiety or depression or to help down-regulate arousal. As requested, we have added the medications that were used as part of the children’s treatment in the cases. The material has been added as a table.

The authors should cite the following cite and papers and discussed them.

Thank you very much for providing us with these excellent references. We have added the relevant ones to the manuscript—the introduction section—and we have kept the others for use as needed in the clinic.

  • https://www.gov.uk/government/publications/covid-19-vaccination-resources-for-children-and-young-people

We have added a sentence about resources for the family and added this reference there.

  • https://www.gov.uk/government/publications/universal-vaccination-of-children-and-young-people-aged-12-to-15-years-against-covid-19/universal-vaccination-of-children-and-young-people-aged-12-to-15-years-against-covid-19

We have added this reference to the section about weighing risks and benefits.

  • https://www.medrxiv.org/content/10.1101/2022.03.22.22272775v1 (https://www.medrxiv.org/content/10.1101/2022.03.22.22272775v1.full.pdf)

We have added this reference to the section about weighing risks and benefits.

  • https://www.ons.gov.uk/peoplepopulationandcommunity/healthandsocialcare/causesofdeath/articles/covid19vaccinationandmortalityinyoungpeopleduringthecoronaviruspandemic/2022-03-22

The previous article covers the topic, and this report is much longer and less pithy.

  • Piccini, B., Pessina, B., Pezzoli, F., Casalini, E., & Toni, S. (2022). COVID-19 vaccination in adolescents and young adults with type 1 diabetes: Glycemic control and side effects. Pediatric diabetes, 23(4), 469–472. https://doi.org/10.1111/pedi.13326

Our cohort did not have any issues with diabetes, so we have not included this reference.

  • Ziv, A., Heshin-Bekenstein, M., Haviv, R., Kivity, S., Netzer, D., Yaron, S., Schur, Y., Egert, T., Egert, Y., Sela, Y., Hashkes, P. J., & Uziel, Y. (2022). Effectiveness of the BNT162b2 mRNA COVID-19 Vaccine among Adolescents with Juvenile-onset Inflammatory Rheumatic Diseases. Rheumatology (Oxford, England), keac408. Advance online publication. https://doi.org/10.1093/rheumatology/keac408

This is a very interesting paper, but our cohort does not have juvenile arthritis. We have not included this particular paper.

  • Lawson-Tovey, S., Machado, P. M., Strangfeld, A., Mateus, E., Gossec, L., Carmona, L., Raffeiner, B., Bulina, I., Clemente, D., Zepa, J., Rodrigues, A. M., Mariette, X., Hyrich, K. L., & EULAR COVAX (2022). SARS-CoV-2 vaccine safety in adolescents with inflammatory rheumatic and musculoskeletal diseases and adults with juvenile idiopathic arthritis: data from the EULAR COVAX physician-reported registry. RMD open, 8(2), e002322. https://doi.org/10.1136/rmdopen-2022-002322

This is a very interesting paper, but our cohort does not have juvenile arthritis or musculoskeletal diseases. We have not included this particular paper.

  • Beckley, M., Olson, A. K., & Portman, M. A. (2022). Tolerability of COVID-19 Infection and Messenger RNA Vaccination Among Patients With a History of Kawasaki Disease. JAMA network open, 5(8), e2226236. https://doi.org/10.1001/jamanetworkopen.2022.26236

This is a very interesting paper but our cohort does not have Kawasaki disease or any other autoimmune pathology.  

Reviewer 4 Report

This is a reasonably good paper but there are weaknesses and, maybe, room for improvement,

1) The symptoms may be temporary and go away in the long term just as symptoms appear temporarily for non-FND candidates upon vaccination. This issue is not addressed in the paper. Perhaps, there should have been a follow-up after a few months to get a sense of the long term. Do the authors have data pertaining to this? If not, the authors should state this as a weakness and suggest ways to improve the research.

2) Another weakness is that the research is unable to specify the specific symptoms that are more manifested upon vaccination. Perhaps. it is because of the small sample size. Again, this needs to be addressed.

3) Statistics has been used but I am unable to find the methodology used or the software used,

Author Response

Reviewer #4

This is a reasonably good paper but there are weaknesses and, maybe, room for improvement,

1) The symptoms may be temporary and go away in the long term just as symptoms appear temporarily for non-FND candidates upon vaccination. This issue is not addressed in the paper. Perhaps, there should have been a follow-up after a few months to get a sense of the long term. Do the authors have data pertaining to this? If not, the authors should state this as a weakness and suggest ways to improve the research.

The reviewer may have missed this, but the information about the length of the relapse or exacerbation is included in cases 1–4. This information is not currently available for case 5 or 6 because they are still unwell at the time of writing this paper. We will, of course, follow them up, but the information is not currently available (they are still unwell as we write). We have added the lack of follow-up data for cases 5 and 6 in the limitations.

2) Another weakness is that the research is unable to specify the specific symptoms that are more manifested upon vaccination. Perhaps. it is because of the small sample size. Again, this needs to be addressed.

The specific symptoms for each child are described in detail in each case. But the reviewer is quite right, the sample is too small to discuss what FND symptoms are the most common ones in response to vaccination. We have also added this as a limitation in the limitations section.

3) Statistics has been used but I am unable to find the methodology used or the software used,

The reviewer #4 may have missed it, but in the section titled, “Analysis of clinical characteristics and self-report data”, we had stated that “Chi-square analyses and independent t-tests were used to calculate differences between the FND and control groups on, respectively, categorical and continuous variables”.

We have added that we used SPSS Statistics 26.

Reviewer 5 Report

This interesting paper describes a cohort of young people diagnosed with functional neurologic disorders who received COVID19 vaccines, comparing them to a group of healthy controls. The observations are intriguing.  However, several aspects might be strengthened

1.     As the authors point out, this study is uninformative about the frequency with which vaccination affects FND in young people. The population studied is highly selected in that patients were initially carefully evaluated neurologically, identified as having functional neurologic disorders and then referred to the Mind-Body Program. Each of these introduces an important selection bias. As discussed by the authors, only a systematic prospective population study could specifically address this question

2.     I’m not sure I understand why these controls were included.  While they help identify some factors predisposing young people to FNDs, the topic of the paper is “COVID19 vaccination in young people with FND”. How is this background information relevant to this question?

3.     Given that only 6 of 61 subjects with FND exhibited any suspected worsening of their disorder after vaccination, it is difficult to reach any conclusions about a role of vaccination.  Are there quantitative data on exacerbations of FND with other stressors, for comparison?   There is a lot of qualitative discussion of this, but if there are data, I must have missed it.

4.     It would have been interesting to determine if there were any differences between the 6 patients who developed new or worsening FND and the 55 who did not. The small numbers make it unlikely any compelling conclusions could be reached but this was the question one would expect a paper with this title to address.

5.     Lesser points:

a.     Patients 5 and 6 received the Pfizer vaccine. Is the vaccine used in the other 4 known?  Are the relative proportions of the 2 vaccines in this cohort reflective of the proportions in the broader population?

b.     Much of the neurobiology described is fascinating but quite speculative.  It would be helpful to emphasize this more clearly.

c.     Narcolepsy (Case 6).  Narcolepsy is not a FND; diagnostic criteria are quite specific and it is almost always HLA linked.  Given the strong association of one specific H1N1 flu vaccine with this disorder, this is a very important question. The authors should be very explicit about the basis of this diagnosis in this subject – and if the diagnosis is accurate, probably remove her from the list of FNDs.

d.     Minor biologic point: current thinking is that there is 1 population of microglia that originates in the CNS and another that arises peripherally

Author Response

Reviewer 5

This interesting paper describes a cohort of young people diagnosed with functional neurologic disorders who received COVID19 vaccines, comparing them to a group of healthy controls. The observations are intriguing.  However, several aspects might be strengthened

  1. As the authors point out, this study is uninformative about the frequency with which vaccination affects FND in young people. The population studied is highly selected in that patients were initially carefully evaluated neurologically, identified as having functional neurologic disorders and then referred to the Mind-Body Program. Each of these introduces an important selection bias. As discussed by the authors, only a systematic prospective population study could specifically address this question.

Yes Reviewer #5 is correct. The study gives us very interesting case-level data, but a much larger population sample is needed to provide information about the frequency with which vaccination affects FND in young people.

Notwithstanding, this case-control study done in a naturalistic setting in a high-risk group does provide very interesting and important data. It shows that even in this high-risk group, the likelihood of developing FND relapse of exacerbation in response to COVID-19 vaccination is very small. This finding is significant for clinicians working with the patient population and discussing vaccination with patients.

The study is also important because it provides clinical case-based data. In the clinic, paediatricians need to be aware of research data from multiple system levels: the statistical population level, which provides information about probabilities, and the individual case level, which provides information about the patients’ lived experience. Knowledge from both system levels is needed to practice paediatrics in a holistic (biopsychosocial) manner.

  1. I’m not sure I understand why these controls were included.  While they help identify some factors predisposing young people to FNDs, the topic of the paper is “COVID19 vaccination in young people with FND”. How is this background information relevant to this question?

Thank you for asking this important question. The controls are very important. They allow us to examine two key predisposing factors that typify most FND research cohorts: increased adverse childhood experiences (ACEs) and increased subjective distress. Without the control group any numbers pertaining to these two factors would be meaningless because there would be nothing to compare them to. 

ACEs. A key feature of young people with FND includes a developmental history characterised by an increased number of adverse childhood experiences. Whilst the FND itself can be triggered by a relatively minor trigger—an emotional event, an injury (fall), or a medical procedure (including vaccination)—this typically occurs on the backdrop of ACEs that are cumulative. The controls allow us to examine this predisposing factor in the current group.

Subjective distress. Another feature of young people with FND is their increased level of distress, presumably related to the increased number of ACEs. The controls allow us to examine this predisposing factor in the current group.

The results of this analysis are reported under the subheading, “Self-report measures and heart rate on admission to the research program”

On admission to the research program, young people in the FND group reported a greater number of ACEs (total ELSQ score: range, 0–10; mean=3.58, SD=2.80; median=2.00) compared to healthy controls (range, 0–3; mean=0.52, SD=0.78; median=0.61) (t(70.57) = 8.22; p<.001)) (see Table 1). On admission to the research program, young people in the FND group also reported higher levels of distress (total DASS score: range, 3–52; mean=27.68, SD=12.86; median=28) compared to healthy controls (range, 0–30; mean=5.72, SD=5.67; median=4.00) (t(85.50)=11.82; p<.001)).

These data about predisposing factors allow us to think about FND triggered by vaccination in a broader biopsychosocial context.

  1. Given that only 6 of 61 subjects with FND exhibited any suspected worsening of their disorder after vaccination, it is difficult to reach any conclusions about a role of vaccination.  Are there quantitative data on exacerbations of FND with other stressors, for comparison?   There is a lot of qualitative discussion of this, but if there are data, I must have missed it.

This is a good question. In a recent paper, we reported outcome pathways—including a pattern of relapse with subsequent stress—in three different cohorts.

Kozlowska, K., et al. (2021). "Psychologically informed physiotherapy as part of a multidisciplinary rehabilitation program for children and adolescents with functional neurological disorder: Physical and mental health outcomes." Journal of Paediatrics and Child Health 57(1): 73-79.

We have added this material to the discussion.

  1. It would have been interesting to determine if there were any differences between the 6 patients who developed new or worsening FND and the 55 who did not. The small numbers make it unlikely any compelling conclusions could be reached but this was the question one would expect a paper with this title to address.

This is a good question. Reviewer #5 is correct, however, in pointing out that the numbers are too small to enable us to look at this issue in a statistical sense.

  1. Lesser points:
  2. Patients 5 and 6 received the Pfizer vaccine. Is the vaccine used in the other 4 known?  Are the relative proportions of the 2 vaccines in this cohort reflective of the proportions in the broader population?

All patients in the vignettes received Pfizer vaccines because this was the only vaccine that was available to them—for their age group—at the time of vaccination. The Moderna vaccine became available some time later (in NSW, Australia).

We have added this information to the text.

  1. Much of the neurobiology described is fascinating but quite speculative.  It would be helpful to emphasize this more clearly.

We have done this as requested. The discussion of probable neurobiology is very important. It provides the reader with an update about what we now know about FND—in the context of two decades of renewed research using novel methodologies that were not previously available. Having this information at ones fingertips will enable clinicians to access it when they have patients who present with FND following vaccination.

Interestingly, patients are very pleased to be given an explanation even if the doctor emphasizes that the explanation is our best current hypothesis based on recent research. It is the lack of an explanation—even hypothetical—that creates distress and anxiety for patients.

  1. Narcolepsy (Case 6).  Narcolepsy is not a FND; diagnostic criteria are quite specific and it is almost always HLA linked.  Given the strong association of one specific H1N1 flu vaccine with this disorder, this is a very important question. The authors should be very explicit about the basis of this diagnosis in this subject – and if the diagnosis is accurate, probably remove her from the list of FNDs.

Thank you for raising this issue and our lack of clarity. This child (case 6) was particularly interesting because vaccination triggered three different illness processes: atypical narcolepsy (narcolepsy type 2), a functional gut disorder, and an FND disorder (with a persistent loss of motor function in the legs and functional/dissociative seizures that changed in semiology over time [a typical feature of functional seizures]).

The diagnosis of narcolepsy (type 2)—what we termed atypical narcolepsy to the family—was made by a paediatric sleep physician based on the clinical presentation, polysomnography, and the multiple sleep latency test. We have put the details of the three diagnoses in the vignette. All involved physicians—the admitting paediatrician, the sleep physician, the gastroenterologist, and the neurologist—agreed that the child had three different illness processes. Importantly, the child’s FND symptoms have made the clinical picture very complex and excluded the possibility of clearly diagnosing cataplexy. In this context, she was diagnosed with narcolepsy type 2.

At the current point in time, the narcolepsy, the functional gut disorder (improved), and the FND are all ongoing. The narcolepsy is being followed up by the sleep team, the functional gut disorder by the family doctor, and the FND by the local physiotherapy service and the local CAMHS (child and adolescent mental health service).

There is a very nice article “just out” that highlights that the differences in children’s presentations affect how the diagnosis is made in paediatrics, and how it is treated. We have amended the Text Box to add information from this article. It is much clearer than previous literature.

Chung, I.-H.; Chin,W.-C.; Huang, Y.-S.;Wang, C.-H. Pediatric Narcolepsy—A Practical Review. Children 2022, 9, 974.

  1. Minor biologic point: current thinking is that there is 1 population of microglia that originates in the CNS and another that arises peripherally

Yes, I think that is right. There are some nice papers by Ginhoux that track this interesting piece of research. It is one of the medical mysteries that has taken some time to untangle.

Round 2

Reviewer 3 Report

This manuscript is too long. Should shorten it.

Please move some of the data to the Supplementary data.

Author Response

This manuscript is too long. Should shorten it.

The manuscript has been shortened: the text about the vaccine rollout in Australia has been moved into the Supplementary Materials. The material about measures and medications has also been moved into the supplementary materials.

Please move some of the data to the Supplementary data.

As noted, the Text Box describing measures and Table with medication history data has been moved into the supplementary materials.

Extensive editing of English language and style required

We have checked and corrected the spelling using Word’s spellchecker (UK English) and PerfectIt 5 (a professional copyediting program).

Author SS has extensive professional editing experience. He has gone over the manuscript. We wonder whether Reviewer #3 may have mistakenly checked the item suggesting that “extensive further editing is required” versus the item suggesting a “spell check”.

Reviewer 4 Report

Improvements seen

Author Response

English language and style are fine/minor spell check required

Improvements seen

We have checked and corrected the spelling using Word’s spellchecker (UK English) and PerfectIt 5 (a professional copyediting program).

Reviewer 5 Report

Although many of my comments were not addressed I suspect the current version is the best that can be done.  I have just 1 minor comment. In text box 6 recognized environmental factors leading to narcolepsy include very specific vaccines, not vaccines generally

Author Response

English language and style are fine/minor spell check required

We have checked and corrected the spelling using Word’s spellchecker (UK English) and PerfectIt 5 (a professional copyediting program).

Although many of my comments were not addressed I suspect the current version is the best that can be done.  I have just 1 minor comment. In text box 6 recognized environmental factors leading to narcolepsy include very specific vaccines, not vaccines generally.

Thank you for picking this up. We have changed the sentence to read, “Recognized environmental factors include infection (strep pyogenes, influenza, H1) and the H1N1 (Pandemrix, GSK) vaccine”.

Round 3

Reviewer 3 Report

This manuscript is too long.

Please move all Tex Boxes to Supplementary data.

Author Response

As requested by Reviewer #3 the Text Boxes have been removed to the supplementary materials.